# Synthesis and Properties of Plasma-Polymerized Methyl Methacrylate via the Atmospheric Pressure Plasma Polymerization Technique

**DOI:** 10.3390/polym11030396

**Published:** 2019-02-28

**Authors:** Choon-Sang Park, Eun Young Jung, Hyo Jun Jang, Gyu Tae Bae, Bhum Jae Shin, Heung-Sik Tae

**Affiliations:** 1School of Electronics Engineering, College of IT Engineering, Kyungpook National University, Daegu 41566, Korea; purplepcs@ee.knu.ac.kr (C.-S.P.); eyjung@knu.ac.kr (E.Y.J.); bs00201@knu.ac.kr (H.J.J.); doctor047@knu.ac.kr (G.T.B.); 2Department of Electronics Engineering, Sejong University, Seoul 05006, Korea; hahusbj@sejong.ac.kr

**Keywords:** methyl methacrylate, encapsulation, atmospheric pressure plasma, plasma polymerization, plasma-polymerized methyl methacrylate (pPMMA), time of flight-secondary ion mass spectrometry (ToF-SIMS), hydrophobicity

## Abstract

Pinhole free layers are needed in order to prevent oxygen and water from damaging flexible electrical and bio-devices. Although polymerized methyl methacrylate (polymethyl methacrylate, PMMA) for the pinhole free layer has been studied extensively in the past, little work has been done on synthesizing films of this material using atmospheric pressure plasma-assisted electro-polymerization. Herein, we report the synthesis and properties of plasma-PMMA (pPMMA) synthesized using the atmospheric pressure plasma-assisted electro-polymerization technique at room temperature. According to the Fourier transform infrared spectroscopy (FT-IR), X-ray photoelectron spectroscopy (XPS), and time of flight-secondary ion mass spectrometry (ToF-SIMS) results, the characteristic peaks from the pPMMA polymer chain were shown to have been detected. The results indicate that the percentage of hydrophobic groups (C–C and C–H) is greater than that of hydrophilic groups (C–O and O–C=O). The field emission-scanning electron microscope (FE-SEM) and thickness measurement results show that the surface morphology is quite homogenous and amorphous in nature, and the newly proposed pPMMA film at a thickness of 1.5 µm has high transmittance (about 93%) characteristics. In addition, the results of water contact angle tests show that pPMMA thin films can improve the hydrophobicity.

## 1. Introduction

Plasma-assisted electro-polymerization is a new research field on the interaction of plasma with monomer and electrolyte solution, and it uses plasma to drive the chemical reactions [1,2,3,4,5,6]. Electrochemically controlled formation of thin polymer films using plasma is one of the feasible routes to design new functional polymer materials. The creation of new polymer materials would require novel technological developments. Recently, there have been a number of interesting reports concerning an aerosol-through-plasma (ATP) technique for generating particulate materials [7,8,9,10,11,12,13,14]. In brief, this technology consists of passing aerosols through various plasma systems, particularly radio frequency (RF) and microwave, to create particulates, nanoparticles, and thin films. Among various plasma systems, by using a nonthermal atmospheric pressure plasma (APP) as a gaseous electrode, plasma-assisted electro-polymerization has attracted increasing attention in the green synthesis of polymer thin films and nanomaterials. Thanks to the low-temperature and cost-effective dry process, the APP polymerization process is a promising growth method for synthesizing polymer thin films with functional features, therefore being well-suited for versatile substrates and applications. The APP polymerization process can obtain high-quality polymer films and nanoparticles, with easily controllable deposition rate, size, and is well-suited for a wide range of substrates. Furthermore, APP reactors do not require a vacuum atmosphere and special equipment. In spite of intensive studies, only a few groups have reported successful plasma polymerization using APP [15,16,17,18,19,20,21,22,23,24,25]. Our previous works recently reported the stationary APP jets (APPJs) polymerization employing the new types of impinging technique and ATP system [26,27,28,29,30]. When discharge is initiated, the flowing inert argon gas can inject electrons and ions into the vaporized monomer bubbles. This impinging technique and ATP system increase the charged particles and density by means of plasmas in the monomer fragmentation (or nucleation) and recombination regions. In this case, the polymer films can have different properties depending on the types of precursors and plasma conditions. Consequently, the advanced plasma-based process for synthesizing new materials including polymers requires a deeper understanding of plasma-assisted electro-polymerization.

Recently, polymerized methyl methacrylate (polymethyl methacrylate, PMMA) having pinhole free characteristics has been widely used as organic thin films, encapsulations, biosensors, and electronics materials [21,22,23,24,25,31,32,33,34,35,36]. The plasma polymerizations of porous conducting polymers, such as aniline, pyrrole, and thiophene, have been successfully implemented using our APP polymerization technique [26,27,28,29,30]. However, there is no report on the synthesis of plasma-PMMA (pPMMA) with pinhole free and nonporous characteristics as a dielectric polymer through plasma-assisted electro-polymerization with the novel APP polymerization technique. Accordingly, this study uses Fourier transform-infrared spectroscopy (FT-IR), X-ray photoelectron spectroscopy (XPS), time of flight-secondary ion mass spectrometry (ToF-SIMS), field emission-scanning electron microscopy (FE-SEM), and transmittance measurement to analyze newly proposed pPMMA film synthesized by APPJs polymerization. The resultant surface wettability is also investigated. 

## 2. Experimental and Characterizations

### 2.1. Atmospheric Pressure Plasma Synthesis and Materials

A stationary atmospheric pressure plasma (APP) polymerization technique by using liquid monomer was conducted by an APP jets (APPJs) and aerosol-through-plasma system mentioned in our previous work [26]. In the previous work, a newly proposed guide tube and a bluff body with impinging jet systems were introduced to suppress the quenching phenomenon from ambient air, thereby increasing the charged particles and plasma energies in the nucleation area [27]. In the case of conventional APPJs without the proposed guide tube and buff body, the plasma was only produced within the area of the three array jets because of the directional characteristics of the streamer-like discharges. In the case of the newly proposed guide tube and bluff body with impinging jet systems, the plasma produced in the impinging region was changed to a broadened glow-like discharge, thereby increasing the plasma region about 60-fold and the deposition area [27,29]. The proposed APPJs with impinging technique and ATP system can produce a broadened and intense plasma discharge with large area deposition and obtain high-quality polymers, with easily controllable morphology, deposition rate, deposition size, and is well-suited for a wide range of substrates. The detailed APP polymerization system, featuring the guide tube and bluff body, employed in this work, was described in the references of [26,27,28,29,30]. In the case of a low gas flow rate below 2500 standard cubic centimeters per minute (sccm) for the main discharge gas and below 300 sccm for vaporizing or carrier monomer gas flow rates, intense or stable or broadened plasma was not produced even though the proposed guide tube and bluff body with impinging jet systems was used [29]. In this experiment, therefore, the argon (Ar) gas was the main plasma discharge gas where its purity was high (99.999%) and its flow rate was fixed at 2500 sccm. Further Ar gas was supplied to the glass bubbler for vaporizing the liquid methyl methacrylate (MMA) monomer (Sigma-Aldrich Co., St. Louis, MO, USA, *M_w_* = 100 g·mol^−1^) at a fixed flow rate of 300 sccm. For plasma generation, a sinusoidal voltage with a peak value of 12 kV and a frequency of 30 kHz were fixed and applied to the proposed APP polymerization device. The used substrates were glasses and polyethylene terephthalates (PETs). The pPMMA was directly deposited on substrates at room temperature by the novel APP polymerization technique. During the APP polymerization experiments, the plasma jet was not moved, and the as such experiments were conducted in stationary deposition. 

### 2.2. Fourier Transform-Infrared Spectroscopy Analysis

The Fourier transform infrared spectra (FT-IR) were used to determine the chemical changes of pPMMA introduced by the APP and measured on a Perkin-Elmer Frontier spectrometer (PerkinElmer, Waltham, MA, USA) between 650 and 4000 cm^−1^. 

### 2.3. X-ray Photoelectron Spectroscopy Analysis

An X-ray photoelectron spectroscopy (XPS) (ESCALAB 250XI surface analysis system, Thermo Fisher Scientific, Waltham, MA, USA) was used to investigate the surface chemical compositions and atomic concentration of the pPMMA films. In the XPS measurement, the voltage and current of the monochromatic Al Kα X-ray source (hv = 1486.7 eV) were 15 kV and 20 mA, respectively. The measurement angle was 60° and the measurement depth estimated to range from 8 to 10 nm. The measurement area was 500 μm × 500 μm and the pressure was about 10^−8^ Pa. The C 1s spectrum (285.0 eV) was used to calibrate the energy scale. Elements present on the deposited surfaces were identified from XPS survey scans and quantified with Thermo Avantage software (v.5.977, Waltham, MA, USA) using a Shirley background and applying the relative sensitivity factors provided by the manufacturer of the instrument. The relative sensitivity factors of C 1s and O 1s were 1.0 and 2.8, respectively. For high-resolution spectra, the constant analyzer energy modes were used at 200 eV for the survey scan and 50 eV pass energy for the element scan, respectively. Since the pPMMA samples and substrates were insulators, we used an additional electron gun to allow for surface neutralization to adjust the charge compensation during the measurements. To curve fit the high-resolution C 1s and O 1s peaks, the deconvolution of C 1s and O 1s peaks was analyzed by the Thermo Avantage software. The peaks were deconvoluted using Gaussian–Lorentzian peak shapes (constrained between 80% and 100% Gaussian) and the full-width at half maximum (FWHM) of each line shape was constrained between 2.0 and 3.0 eV.

### 2.4. Time of Flight-Secondary Ion Mass Spectrometry Analysis

The surface structure and composition of the pPMMA films were examined by the time of flight-secondary ion mass spectrometry (ToF-SIMS) V instrument (ION-TOF GmbH, Munster, Germany) with a bismuth primary-ion (Bi_3_^+^) gun source. The pressure in the ToF-SIMS chamber was maintained at less than 1 × 10^−9^ Torr. Bi_3_^+^ (0.5 pA) accelerated at 30 keV was used as the analysis (primary) gun. The negative-ion and positive-ion mass spectra of a 500 μm × 500 μm area were acquired at a Bi_3_^+^ primary-ion beam of 30 keV. For a more accurate mass scale of pPMMA film, the spectra were calibrated based on the characteristic pPMMA ion, C_4_H_5_O_2_^−^ including the known CH^−^ and C_4_H^−^ ions. 

### 2.5. Surface Morphology Study

The surface morphology of the pPMMA films on the glass substrates was characterized using a field emission-scanning electron microscope (FE-SEM) (Hitachi SU8220, Hitachi, Tokyo, Japan) with accelerating voltage and current of 3 or 5 kV and 10 mA, respectively. The samples for FE-SEM were conductive platinum-coated before loading into the chamber. 

### 2.6. Thickness Measurement

The thickness of pPMMA was determined using a KLA Tencor P-7 stylus profilometer (Milpitas, CA, USA) with a diamond probe having a radius of 2 μm. 

### 2.7. Optical Transmittance

The optical transmittance (or transmission) of pPMMA film was performed on a glass substrate followed by measurement of its transmission by using of UV–Vis spectrophotometer (Cary 5G, Varian). The spectrum ranged from 300 to 800 nm with a scanning step of 0.5 nm. Glass and PET plates were used as reference. The pPMMA samples were used for measurement of optical transmittance and the results were expressed as a mean value.

### 2.8. Water Contact Angle Measurements

The sessile drop method using Drop Shape Analyzers (KRUSS, DSA100, Germany) was used to measure the surface wettability of the pPMMA films grown on the glass and PET substrates. After the ultrapure water droplet contacted the sample surfaces, the contact angles were measured by the built-in charge-coupled device (CCD) video camera with the software provided by the manufacturer. The average value of the contact angle was taken after five measurements for each sample. 

## 3. Results and Discussion

The FT-IR absorption spectrum of the newly proposed pPMMA film grown on glass substrate when using the APP polymerization technique after 90 min deposition is shown in Figure 1. Bands from 3000 to 3600 cm^−1^ of the OH bonding group, implying oxidation of the particles induced by quenching from ambient air, were not observed thanks to using the newly proposed guide tube and bluff body systems. The FT-IR spectra of the pPMMA polymer were found to exhibit absorption bands at 2989 and 2951 cm^−1^. These bands were caused by –CH_3_ asymmetric stretching. A sharp band located at 1731 cm^−1^ was ascribed to the carbonyl group. The pPMMA showed an IR absorption band at 1433 cm^−1^ due to the asymmetric bending vibration (CH_3_) of the methyl group. The peak at 1381 cm^−1^ was attributed to OCH_3_ deformation of pPMMA. The characteristic absorption bands at 1265 and 1145 cm^−1^ respectively correspond to C–O–C stretching and the C–O group of pristine PMMA polymer. The band at 1193 cm^−1^ was due to –OCH_3_ stretching. The absorption bands at 977 and 716 cm^−1^ were assigned to the CH_2_ wagging and rocking modes of pPMMA, respectively [37,38,39,40]. 

The XPS survey spectra, with elemental composition percent (Figure 2a, insets) of the atomic distribution and detailed high-resolutions of C 1s and O 1s spectra in the newly proposed pPMMA thin films grown on glass substrate when using the APP polymerization technique after 90 min deposition, are shown in Figure 2. The element content was calculated by the ratio of the corresponding integral area of the peaks. The XPS survey spectra in Figure 2a are typically observed in the signals corresponding to C 1s (285.0 eV) and O 1s (531.0 eV) electronic orbitals of pPMMA [41,42]. The C and O atoms belong to the MMA monomer structure. To get further insight into the types and ratios of the surface functional groups of the pPMMA, the curve fitting of the high-resolution C 1s and O 1s peaks was analyzed by the XPS peak. The assignments of the fitted components and the compositions of the C 1s and O 1s levels are summarized in Table 1. The C 1s peaks of pPMMA were decomposed by curve fitting into three components; the component at 285.0 eV corresponding to C–C and C–H bonds, the component at 286.8 eV corresponding to C–O bands, and the component at 289.1 eV corresponding to O–C=O bonds. The O 1s peaks of pPMMA can be divided into two component peaks, which respectively indicate two oxygen-containing groups; C=O (531.0 eV) and C–O (533.2 eV). After fitting the relative intensities of the two peaks, the peak corresponding to the C=O group was shown to be higher than the one of the C–O group in Figure 2c. Comparing the C 1s spectra, the relative intensities of the C–O and O–C=O groups were lower, while the relative intensities of the C–C and C–H groups in the same spectra were higher in Figure 2b. The high peak could be attributed to cross-linking reactions and the formation of new C–C bonds. The values in Table 1 denote the ratio of carbon atoms in the hydrophobic groups (C–C and C–H) to those in the hydrophilic groups (C–O and O–C=O) on the surface. The results indicate that the percentage of hydrophobic groups is greater than that of hydrophilic groups. This means that our pPMMA surface becomes carbon-rich and eventually it is expected to increase the water contact angles (WCAs) after pPMMA coating [41,42].

The pPMMA films were characterized in both negative- and positive-ion modes using the ToF-SIMS method in order to determine the specific structures. The negative-ion and positive-ion spectra (0–200 amu) of ToF-SIMS on the surface of the newly proposed pPMMA thin films grown on a glass substrate when using the proposed APP polymerization technique after 90 min deposition are shown in Figure 3. As shown in Figure 3a using negative-ion mode, the ions at *m/z* = 31, 55, 71, 85, 141, and 185 were assigned to CH_3_O^−^, C_3_H_3_O^−^, C_3_H_3_O_2_^−^, C_4_H_5_O_2_^−^, C_8_H_13_O_2_^−^, and C_9_H_13_O_4_^−^, respectively [43,44,45,46]. These fragment ions arose from the pPMMA chains. The most abundant fragment ions were CH_3_O^−^ and C_4_H_5_O_2_^−^, which were the monomer where the methyl group (CH_3_) was removed from the ether linkage (H_3_C–O–C) [44,45]. The small negative hydroxyl ion OH^−^ (*m/z* = 17) implying the oxidation of the pPMMA thin films induced by quenching from ambient air was observed thanks to using the proposed guide tube and bluff body with impinging jet systems. As shown in Figure 3b using the positive-ion mode, some characteristic peaks from the pPMMA polymer chain were detected. The ions at *m/z* = 15, 27, 31, 39, 41, 55, 59, 69, 77, and 91 were assigned to CH_3_^+^, C_2_H_3_^+^, CH_3_O^+^, C_3_H_3_^+^, C_3_H_5_^+^, C_4_H_7_^+^, C_2_H_3_O_2_^+^, C_4_H_5_O^+^, C_6_H_5_^+^, and C_7_H_7_^+^, respectively. These fragment ions also originated from the pPMMA chains.

The characteristics of the conductor or semiconductor layer can be influenced by the surface morphology of the coated dielectric layer [47,48]. In this sense, the plane and cross-sectional view SEM images of the newly proposed pPMMA thin films grown on glass substrates when using the proposed APP polymerization technique after 90 min deposition are displayed in Figure 4. As shown by the SEM results of Figure 4, the pPMMA film had a deposition rate of about 0.023 μm·min^−1^, and had no pits and pinholes. The surface of the pPMMA film was observed to be homogeneous, amorphous, and smooth, which is considered to be one of the most important features of the dielectric layer required in organic thin films, encapsulations, biosensors, and electronics applications. The SEM images of Figure 4 illustrating the pPMMA films with the self-assembled mesoscopic structure, confirm that the proposed plasma-assisted electro-polymerization technique enables large-area fabrication of mesoscopic structured pPMMA films.

The variation in the film thickness of the newly proposed pPMMA thin films grown on glass substrates when using the proposed APP polymerization technique at different deposition times is shown in Figure 5. Under various deposition times, the deposition rates were about 0.025 μm·min^−1^ for 60 min deposition time and about 0.023 μm·min^−1^ for 90 min deposition time, which was estimated to be low. However, the deposition rates were increased significantly and maintained after 120 min deposition time at about 0.036 μm·min^−1^ in the proposed APP polymerization technique. This means that the deposition rates in our APP polymerization system remain almost constant after 120 min deposition time.

The optical transmittance of pPMMA plays an important role when used as optical material. Therefore, the variation in the optical transmittance (or transmission) of the newly proposed pPMMA thin films grown on glass substrates when using the proposed APP polymerization technique at different deposition times is shown in Figure 6. The pPMMA thin film had a high transparency within the range of visible wavelength (400–700 nm), which could reach up to 93% at 600 nm. As the deposition times were increased, the corresponding transmittances decreased slightly.

To calculate the surface wettability or hydrophobicity of the newly proposed polymer, the WCAs of pPMMA were measured [43,47,48,49]. The WCAs analysis is a simple, rapid, and direct method to evaluate the hydrophobic or hydrophilic feature of a surface. The variation in the WCAs on the pristine (bare) substrates and pPMMA thin films grown on glass and polyethylene terephthalate (PET) substrates when using the proposed APP polymerization technique after 90 min deposition is shown in Figure 7. From Figure 7, after deposition, we can see that the WCAs of the pPMMA thin films gradually increased for both glass and PET substrates. The results of the WCA tests showed that the pPMMA thin films could increase the WCAs after deposition. The increased WCAs of the pPMMA thin films were presumably due to the cleavage of the hydrophilic groups and the newly formed hydrophobic groups (C–C and C–H). In addition, we observed a change in the WCAs depending on the substrates. The surface roughness (root mean square roughness, R_q_) of the pPMMA thin films on the glass substrate was 25.9 nm, whereas the R_q_ of the pPMMA thin films on the PET substrate was 0.6 nm in Appendix A and Appendix A. The WCA of the glass sample decreased as the R_q_ was increased. The changed WCAs of the pPMMA thin films on both glass and PET substrates were due to the differences in the surface roughness [50,51]. It is a typical phenomenon that appears on hydrophilic surfaces depending on the Wenzel theory [52]. The detailed atomic force microscope images and the R_q_ of the pPMMA thin films surface grown on glass and PET substrates are shown in Appendix A and Appendix A. 

## 4. Conclusions

In this work, we used for the first-time APP polymerization techniques to provide further insight into the process of electro-polymerization of MMA monomer. The pPMMA thin films were successfully obtained by the atmospheric pressure plasma-assisted electro-polymerization technique. The experimental results show that the newly proposed pPMMA has quite homogenous, amorphous, pinhole free, and high transmittance characteristics. In addition, the results of the water contact angle tests show that the pPMMA thin films can improve the hydrophobicity. The plasma-assisted electro-polymerization technique presented in this study is very simple, highly reproducible, and permits fabrication of large areas of mesoscopic structured films having potential as membranes, organic, inorganic materials, and photonic molecules.

## Figures and Tables

**Figure 1 polymers-11-00396-f001:**
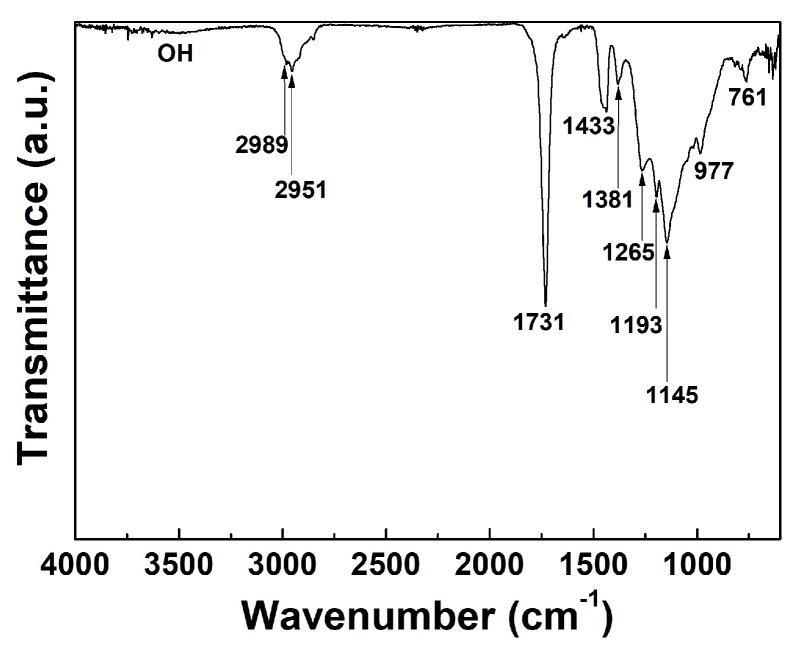
Fourier transform infrared spectroscopy (FT-IR) spectra of plasma-polymerized methyl methacrylate (pPMMA) thin films grown on a glass substrate when using the proposed atmospheric pressure plasma (APP) polymerization technique after 90 min deposition.

**Figure 2 polymers-11-00396-f002:**
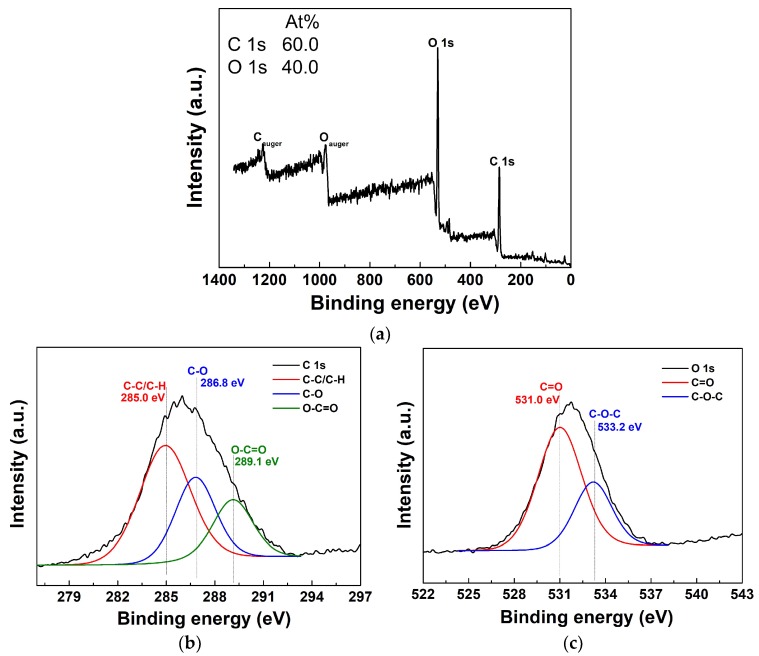
(**a**) X-ray photoelectron spectroscopy (XPS) survey spectra, detailed high-resolution with deconvolutions of (**b**) C 1s, and (**c**) O 1s spectra of pPMMA thin films grown on a glass substrate when using the proposed APP polymerization technique after 90 min deposition. Insets in (a) represent the atom percent in the pPMMA film.

**Figure 3 polymers-11-00396-f003:**
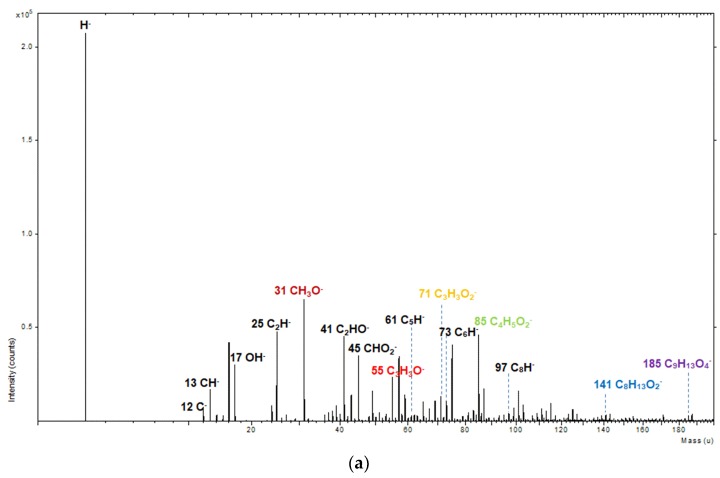
(**a**) Negative-ion and (**b**) positive-ion spectra (0–200 amu) of time of flight-secondary ion mass spectrometry (ToF-SIMS) on the surface of pPMMA thin films grown on a glass substrate when using the proposed APP polymerization technique after 90 min deposition.

**Figure 4 polymers-11-00396-f004:**
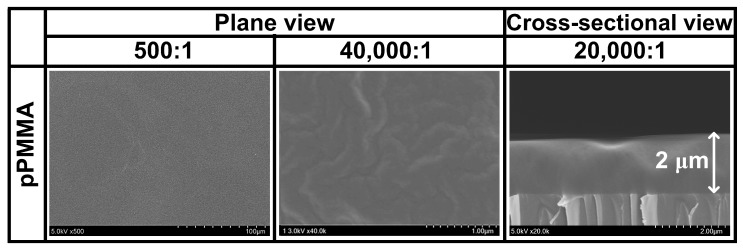
Plane and cross-sectional views of scanning electron microscopy (SEM) images of pPMMA thin films grown on a glass substrate when using the proposed APP polymerization technique after 90 min deposition.

**Figure 5 polymers-11-00396-f005:**
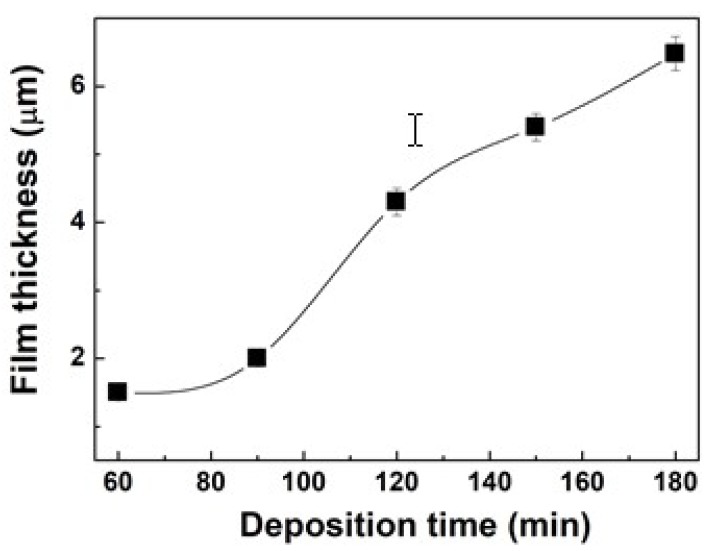
Variation in the film thickness of pPMMA thin films grown on glass substrates when using the proposed APP polymerization technique at different deposition times.

**Figure 6 polymers-11-00396-f006:**
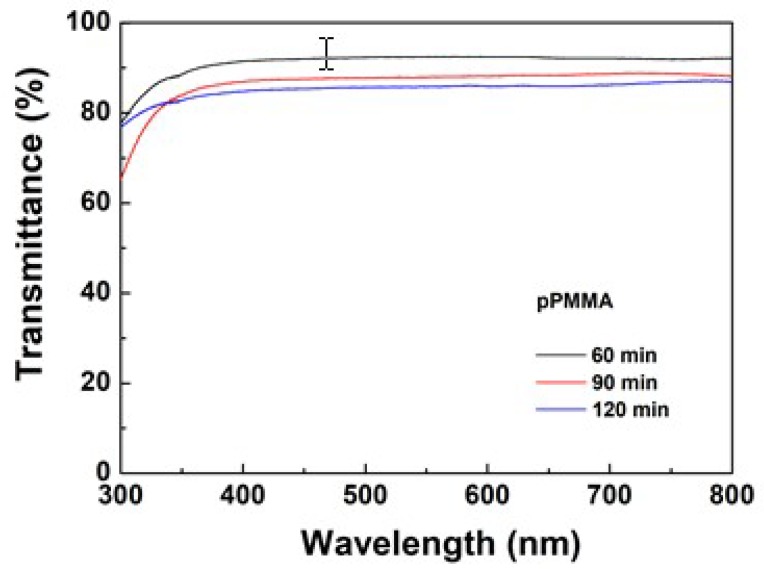
Variation in the optical transmittance (or transmission) of pPMMA thin films grown on glass substrates when using the proposed APP polymerization technique at different deposition times.

**Figure 7 polymers-11-00396-f007:**
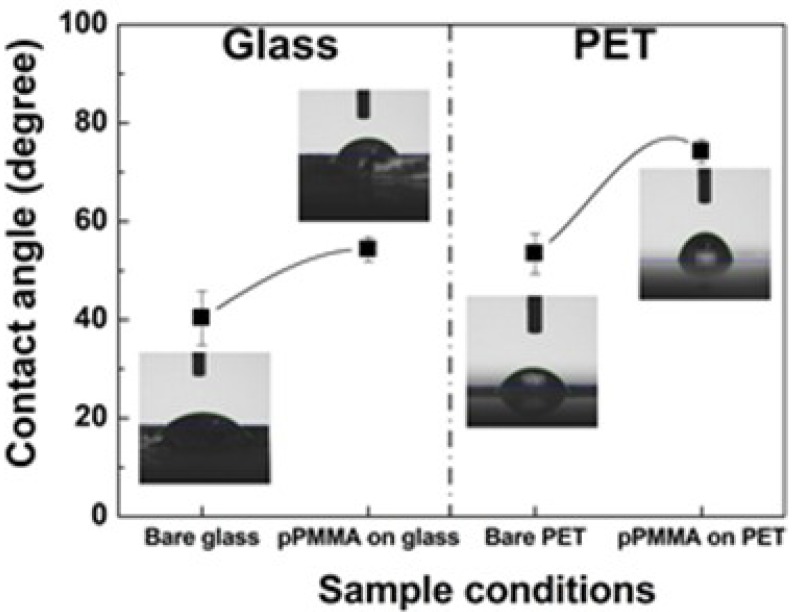
Variation in the water contact angles (WCAs) on the pristine (bare) substrates and pPMMA thin films grown on glass and polyethylene terephthalate (PET) substrates when using the proposed APP polymerization technique after 90 min deposition. Insets represent photographs of the water droplets on the pristine and pPMMA film.

**Table 1 polymers-11-00396-t001:** The contents of functional groups of C 1s and O 1s core level spectra of plasma-polymerized methyl methacrylate (pPMMA) film observed in the XPS spectra of Figure 2b,c.

Sample	Concentrations of Correlative Functional Group (%)				
	C 1s			O 1s	
	C–C/C–H (285.0 eV)	C–O (286.8 eV)	O–C=O (289.1 eV)	C=O (531.0 eV)	C–O (533.2 eV)
pPMMA	54.9	29.4	15.7	68.9	31.1

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
