# Peer review of "Synthesis and Properties of Plasma-Polymerized Methyl Methacrylate via the Atmospheric Pressure Plasma Polymerization Technique"

_polymers, 2019, doi:10.3390/polym11030396_

Round 1
Reviewer 1 Report
The paper from Park and Tae shows a study of plasma polymerization of PMMA using an atmospheric pressure plasma jet. Although the paper present some realistic results, it sounds to me that the literature regarding the many many papers published about PMMA plasma polymerization is lacking. The interest of using a plasma jet is not explained, and the deposition rates are a) low (around 20 nm/min) and b) strange as the deposition rate increases with time. This is not common in controlled plasma polymerization, as the deposition rate should be constant. The quality of the XPS spectra is poor, and the fitting procedure is not explained. This is however crucial as the authors compare the ratio of "polar" groups to "non polar" groups based on XPS. The terms "electropolymerization" and "electrochemistry" are similarly not often used when people deal with plasma polymerization. Some shortcuts are also used : the authors claim that ultimately the surface is expected to be hydrophobic (as it is carbon rich) (see p.4 line 164, but the contact angle p8 , Fig.7 shows a contact angle lower than 90, meaning that the surface is still hydrophilic.. Also, why is the contact angle of PMMA on PET different than the one of PMMA on glass ???? It should be the same, as it is still PMMA ???
Author Response
Revised manuscript number MS #Polymers-414543 entitled “Synthesis and Properties of Plasma-Polymerized Methyl Methacrylate via Atmospheric Pressure Plasma Polymerization Technique” by C.-S. Park et al.
First of all, the authors really appreciate the reviewers’ valuable comments for the paper. Based on the reviewers’ comments, the descriptions for the experimental results are clarified in the revised manuscript. In addition, the related explanations and discussions are compensated and intensified in the revised manuscript. As a result, the X-ray photoelectron spectroscopy (XPS) results in Figure 2 with Table 1, the time of flight-secondary ion mass spectrometry (ToF-SIMS) results in Figure 3, and the film thickness results in Figure 5 are modified in the revised manuscript to express the experimental data more clearly as per the reviewers’ recommendations. In addition, new 13 references [14-24, 50, 51], including the published papers of the group of ‘Morent and De Geyter’, are also provided in the revised manuscript. Total figures, table, and references changed are given as follows.
Old Manuscript | Revised Manuscript |
Fig. 1 | Fig. 1 |
Fig. 2 | Fig. 2 [Modified] |
Fig. 3 | Fig. 3 [Modified] |
Fig. 4 | Fig. 4 |
Fig. 5 | Fig. 5 [Modified] |
Fig. 6 | Fig. 6 |
Fig. 7 | Fig. 7 |
Table 1. | Table 1. [Modified] |
[Ref.] 1, 2, 3, 4, 5, 6, 7, 8, 9, 10, 11, 12, 13 | [Ref.] 1, 2, 3, 4, 5, 6, 7, 8, 9, 10, 11, 12, 13 |
[Ref.] 14, 15, 16, 17, 18, 19, 20, 21, 22, 23, 24 [New] | |
[Ref.] 14, 15, 16, 17, 18, 19, 20, 21, 22, 23, 24, 25, 26, 27, 28, 29, 30, 31, 32, 33, 34, 35, 36, 37, 37, 38 | [Ref.] 25, 26, 27, 28, 29, 30, 31, 32, 33, 34, 35, 36, 37, 37, 38, 39, 40, 41, 42, 43, 44, 45, 46, 47, 48, 49 |
[Ref.] 50, 51 [New] |
I. Upon the reviewer #1’ comments
The paper from Park and Tae shows a study of plasma polymerization of PMMA using an atmospheric pressure plasma jet. |
► We appreciate your kind and detailed assessment of the work presented in this paper.
1. Although the paper present some realistic results, it sounds to me that the literature regarding the many many papers published about PMMA plasma polymerization is lacking. |
► The authors appreciate the in-depth comments from the reviewer. According to the reviewer’ comment, the new 11 references [14-24], including the published papers of the group of ‘Morent and De Geyter’, are newly provided in the revised manuscript. In addition, the related explanations and discussions including introduction are compensated and intensified in the revised manuscript.
At lines 46-50 on page 2 in Introduction:
“The APP polymerization process can obtain a high-quality polymer films and nanoparticles, with easily controllable deposition rate, size, and well-suited for a wide range of substrates. Furthermore, APP reactors do not require a vacuum atmosphere and special equipment. In spite of intensive studies, only a few groups have reported successful plasma polymerization using APP [14-24].”
New References:
[14] Deynse, A. V.; Cools, P.; Leys, C.; Geyter, N. D.; Morent, R. Surface activation of polyethylene with an argon atmospheric pressure plasma jet: Influence of applied power and flow rate. Applied Surface Science 2015, 328, 269-278.
[15] Cools, P.; Sainz-Garcı´, E.; Geyter, N. D.; Nikiforov, A.; Blajan, M.; Shimizu, K.; Alba-Elı´as, F.; Leys, C.; Morent, R. Influence of DBD inlet geometry on the homogeneity of plasma-polymerized acrylic acid films: The use of a microplasma–electrode inlet configuration. Plasma Process and Polymers 2015, 12, 1153-1163.
[16] Vrekhema, S. V.; Cools, P.; Declercq, H.; Tongel, A. V.; Vercruysse, C.; Cornelissen, M.; Geyter, N. D.; Morent, R. Application of atmospheric pressure plasma on polyethylene for increased prosthesis adhesion. Thin Solid Films 2015, 596, 256-263.
[17] Morent, R.; Geyter, N. D.; Trentesaux, M.; Gengembre, L.; Dubruel, P.; Leys, C.; Payen, E. Stability study of polyacrylic acid films plasma-polymerized on polypropylene substrates at medium pressure. Applied Surface Science 2010, 257, 372-380.
[18] Geyter, N. D.; Morent, R.; Vlierberghe, S. V.; Dubruel, P.; Leys, C.; Gengembre, L.; Schacht, E.; Payen, E. Deposition of polymethyl methacrylate on polypropylene substrates using an atmospheric pressure dielectric barrier discharge. Progress in Organic Coatings 2009, 64, 230-237.
[19] Geyter, N. D.; Morent, R.; Leys, C.; Gengembre, L.; Payen, E. Treatment of polymer films with a dielectric barrier discharge in air, helium and argon at medium pressure. Surface & Coatings Technology 2007, 201, 7066-7075.
[20] Casserly, T. B.; Gleason, K. K. Effect of substrate temperature on the plasma polymerization of poly(methyl methacrylate). Chemical Vapor Deposition 2006, 12, 59-66.
[21] Kasih, T. P.; Kuroda, S.-I.; Kubota, H. Poly(methyl methacrylate) films deposited via non-equilibrium atmospheric pressure plasma polymerization using argon as working gas. Plasma Processes and Polymers 2007, 4, 648-653.
[22] Vrekhem, S. V.; Morent, R.; Geyter, N. D. Deposition of a PMMA coating with an atmospheric pressure plasma jet. J. Coat. Technol. Res. 2018, 15(4), 679-690.
[23] Cools, P.; Geyter, N. D.; Vanderleyden, E.; Barberis, F.; Dubruel, P.; Morent, R. Adhesion improvement at the PMMA bone cement-titanium implant interface using methyl methacrylate atmospheric pressure plasma polymerization. Surface & Coatings Technology 2016, 294, 201-209.
[24] Vrekhem, S. V.; Vloebergh, K.; Asadian, M.; Vercruysse, C.; Declercq, H.; Tongel, A. V.; Wilde, L. D.; Geyter, N. D.; Morent, R. Improving the surface properties of an UHMWPE shoulder implant with an atmospheric pressure plasma jet. Scientific Reports 2018, 8, 4720.
2. The interest of using a plasma jet is not explained, and the deposition rates are a) low (around 20 nm/min) and b) strange as the deposition rate increases with time. This is not common in controlled plasma polymerization, as the deposition rate should be constant. |
► We appreciate your valuable comment. First of all, to make clear the interest of using a plasma jet in this study, many sentences and phrases in ‘Introduction’ and ‘Results and Discussion’ are modified and newly provided in the revised manuscript and related new many references [14-24] are also provided in the revised manuscript based on reviewer’ comments.
In addition, under various deposition times, the deposition rates were about 0.025 μm·min-1 for 60 min deposition time and about 0.023 μm·min-1 for 90 min deposition time, which was estimated to be low. However, the deposition rates were increased significantly and maintained after 120 min deposition time at about 0.036 μm·min-1 in proposed APP polymerization technique. Therefore, according to your comment, Fig. 5 with deposition rate changes under more various times (5 point) are modified to express the experimental data more clearly and in order to avoid potential confusion in the revised manuscript, and the following sentences are modified and newly provided in the revised manuscript.
At lines 46-50 on page 2 in Introduction:
“The APP polymerization process can obtain a high-quality polymer films and nanoparticles, with easily controllable deposition rate, size, and well-suited for a wide range of substrates. Furthermore, APP reactors do not require a vacuum atmosphere and special equipment. In spite of intensive studies, only a few groups have reported successful plasma polymerization using APP [14-24].”
At lines 73-95 on pages 2 and 3 in Experimental and Characterizations:
“A stationary atmospheric pressure plasma (APP) polymerization technique by using liquid monomer was conducted by an APP jets (APPJs) and aerosol-through-plasma systems mentioned in our previous work [25]. In the previous, the newly proposed guide tube and bluff body with impinging jet systems were introduced to suppress the quenching phenomenon from ambient air, thereby increasing the charged particles and plasma energies in the nucleation area [26]. The proposed APPJs with impinging technique and ATP system can produce a broadened and intense plasma discharge with large area deposition and obtain high-quality polymers, with easily controllable morphology, deposition rate, deposition size, and well-suited for a wide range of substrates. The detailed APP polymerization system, featuring the guide tube and bluff body, employed in this work, was described in the references of [25-29]. In the case of a low gas flow rate below 2500 standard cubic centimeters per minute (sccm) for main discharge gas and below 300 sccm for vaporizing or carrier monomer gas flow rates, intense or stable or broadened plasma was not produced even though the proposed guide tube and bluff body with impinging jet systems was used [28]. In this experiment, therefore, the argon (Ar) gas was a main plasma discharge gas where its purity was high (99.999%) and its flow rate was fixed at 2500 sccm. Another Ar gas was supplied to the glass bubbler for vaporizing the liquid methyl methacrylate (MMA) monomer (Sigma-Aldrich Co., St. Louis, MO, USA, Mw = 100 g·mol−1) at a fixed flow rate of 300 sccm. For plasma generation, the sinusoidal voltage with a peak value of 12 kV and a frequency of 30 kHz was fixed and applied to the proposed APP polymerization device. The used substrates were glasses and polyethylene terephthalates (PETs). The pPMMA was directly deposited on substrates at room temperature by novel APP polymerization technique. During APP polymerization experiments, the plasma jet was not moved, and as such experiments were conducted in a stationary deposition.”
At lines 253-255 on page 9 in Results and Discussion:
“However, the deposition rates were increased significantly and maintained after 120 min deposition time at about 0.036 μm·min-1 in proposed APP polymerization technique.”
Fig. 5 [Modified]
3. The quality of the XPS spectra is poor, and the fitting procedure is not explained. This is however crucial as the authors compare the ratio of "polar" groups to "non polar" groups based on XPS. |
► We appreciate your kind and detailed assessment of the work presented in this paper. According to the reviewer’ comment, to clearly describe this mechanism, the fitting procedure, program, relative sensitivity factors, and FWHM are newly provided in the revised manuscript, and the high-resolution peaks with deconvolutions of XPS spectrum of Fig. 2 are modified in the revised manuscript.
At lines 101-118 on page 3 in Experimental and Characterizations:
“The X-ray photoelectron spectroscopy (XPS) (ESCALAB 250XI surface analysis system, Thermo Fisher Scientific, Waltham, MA, USA) was used to investigate the surface chemical compositions and atomic concentration of the pPMMA films. In the XPS measurement, the voltage and current of the monochromatic Al Kα X-ray source (hv = 1486.7 eV) was 15 kV and 20 mA, respectively. The measurement angle was 60° and the measurement depth was estimated to range from 8 to 10 nm. The measurement area was 500 μm × 500 μm and the pressure was about 10−8 Pa. The C 1s spectrum (285.0 eV) was used to calibrate the energy scale. Elements present on the deposited surfaces were identified from XPS survey scans and quantified with Thermo Avantage software (v.5.977, Waltham, MA, USA) using a Shirley background and applying the relative sensitivity factors provided by manufacturer of the instrument. The relative sensitivity factors of C 1s and O 1s were 1.0 and 2.8, respectively. For high-resolution spectra, the constant analyzer energy modes were used at 200 eV for survey scan and 50 eV pass energy for element scan, respectively. Since the pPMMA samples and substrates were insulators, we used an additional electron gun to allow for surface neutralization to adjust the charge compensation during the measurements. To curve fit the high-resolution C 1s and O 1s peaks, the deconvolution of C 1s and O 1s peaks was analyzed by the Thermo Avantage. The peaks were deconvoluted using Gaussian–Lorentzian peak shapes (constrained between 80 and 100% Gaussian) and the full-width at half maximum (FWHM) of each line shape was constrained between 2.0 and 3.0 eV.”
(a)
(b) (c)
Fig. 2 [Modified]
4. The terms "electropolymerization" and "electrochemistry" are similarly not often used when people deal with plasma polymerization. |
► The authors appreciate the in-depth comments from the reviewer. According to the reviewer’ comment, to conform the jargon used in plasma community, the words of ‘electrochemical’ and ‘electrochemistry’ are removed in ‘Abstract’ and ‘Introduction’ in the revised manuscript.
At lines 14-17 on page 1 in Abstract:
“Although the polymerized methyl methacrylate (polymethyl methacrylate, PMMA) for pin hole free layer has been studied extensively in the past, little work has been done on synthesizing films of this material using atmospheric pressure plasma-assisted electropolymerization.”
At lines 33-35 on page 1 in Introduction:
“Plasma-assisted electropolymerization is a new research field of the interaction of plasma with monomer and electrolyte solution, and it uses plasma to drive the chemical reactions [1-6].”
At lines 41-43 on page 1 in Introduction:
“Among various plasma systems, by using a nonthermal atmospheric pressure plasma (APP) as a gaseous electrode, plasma-assisted electropolymerization has attracted increasing attention in the green synthesis of polymer thin films and nanomaterials.”
At lines 55-56 on page 2 in Introduction:
“In this case, the polymer films can have the different properties depending on the types of precursors and plasma conditions.”
5. Some shortcuts are also used : the authors claim that ultimately the surface is expected to be hydrophobic (as it is carbon rich) (see p.4 line 164, but the contact angle p8 , Fig.7 shows a contact angle lower than 90, meaning that the surface is still hydrophilic. |
► We appreciate your kind and detailed assessment of the work presented in this paper. According to the reviewer’ comment, the use of ‘hydrophobic’ are changed to ‘increase of WCA’, and the related sentences are modified in the revised manuscript in order to avoid potential confusion.
At lines 189-191 on page 5 in Results and Discussion:
“This means that our pPMMA surface becomes carbon-rich and eventually it is expected to increase the water contact angles (WCAs) after pPMMA coating [40,41].”
At lines 269-280 on page 11 in Results and Discussion:
“To calculate the surface wettability or hydrophobicity of the newly proposed polymer, WCAs of pPMMA were measured [46-49]. The WCAs analysis is a simple, rapid, and direct method to evaluate the hydrophobic or hydrophilic feature of a surface. The variation in the WCAs on the pristine (bare) substrates and pPMMA thin films grown on glass and polyethylene terephthalate (PET) substrates when using proposed APP polymerization technique after 90 min deposition is shown in Figure 7. From Fig. 7, after depositon, we can see that the WCAs of the pPMMA thin films were gradually increased for both glass and PET substrates. The results of WCA tests showed that the pPMMA thin films could increase the WCAs after deposition. The increased WCAs of the pPMMA thin films were presumably due to the cleavage of hydrophilic groups and newly formed hydrophobic groups (C-C and C-H).”
6. Also, why is the contact angle of PMMA on PET different than the one of PMMA on glass ???? It should be the same, as it is still PMMA ??? |
► We appreciate your valuable comment. Although we did not measure the surface roughness of the pPMMA films on the PET and glass substrates, some published papers observed a significant change in the contact angle depending on the film thickness, roughness, and substrate, as seen in 2 references [50, 51]. Consequently, according to the reviewer’ comment, the related sentences are newly provided in the revised manuscript in order to avoid potential confusion. Finally, the authors again thank the reviewer’ efforts for improving the context of this paper.
At lines 280-282 on page 11 in Results and Discussion:
“In addition, we observed a change in the WCAs depending on the substrates. The changed WCAs of the pPMMA thin films on both glass and PET substrates were probably due to the differences in the surface roughness, thickness, and substrates [50,51].”
New References:
[50] Choi, B. K.; Lee, I. H.; Kim J.H.; Chang, Y. J. Tunable wetting property in growth mode-controlled WS2 thin films. Nanoscale Research Letters 2017, 12, 262.
[51] Ahmad, J.; Bazaka, K.; Oelgemöller M.; Jacob, M. V. Wetting, solubility and chemical characteristics of plasma-polymerized 1-isopropyl-4-methyl-1,4-cyclohexadiene thin films. Coatings 2014, 4, 527-552.

Reviewer 2 Report
Dear Authors,
First of all, i would like to say that this manuscript on ppPMMA was interesting to read and well written.
I do have have the following remarks on your manuscript:
- Deposition parameters are missing: power and/or voltage. stationary deposition or moving jet? Especially since only 1 condition was tested in this paper, something which might vary from monomer to momoner in order to get the optimal deposition parameters. This limits the value of this paper.
- XPS the XPS deconvolution is still missing some information (program to do it, relative sensitivity factors, FWHM constraints... )
- XPS results: the high res spectra of your C1s are really poor. I would recommend remeasurements with optimized parameters. In the table, only give 1 digit after the comma.
- FT-IR: it might just be optical, but the peak for OH at 3300 seems rather small compared to what you would expect for PMMA. does this suggest fragmenting? if so, do you see this is TOF SIMS and XPS?
- TOF-SIMS: not enough attention is given to those peaks that cannot be linked to the traditional backbone of PMMA.
- Deposition rate is not linear, which differs from the usual pattern observed for plasma polymerization at a fixed power. How do you explain this? also 3 point measurement is poor, prefer to have 5 points for this.
- There is no stability testing whatsoever. Thickness measurements for 1 day up to 3-4 weeks into aqeous solution would be essential for this paper to be published.
- you do very long deposition times, are
- the group of Morent and De Geyter have done quite a lot of research on PMMA like coatings using plasma in the last 4 years, including APPJ polymerization, yet none of these papers are mentioned. (Look for van Vrekhem and Cools)
Finally, in my opinion, the introduction is a bit misleading. you are using terms such as electropolymerization or electrochemical, which are not common in the plasma community. Therefore I believed that you were doing something new while it is a continuation of what has been done before. I would recommend then to rewrite the intro and abstract conform the jargon used in our community.
Author Response
Revised manuscript number MS #Polymers-414543 entitled “Synthesis and Properties of Plasma-Polymerized Methyl Methacrylate via Atmospheric Pressure Plasma Polymerization Technique” by C.-S. Park et al.
First of all, the authors really appreciate the reviewers’ valuable comments for the paper. Based on the reviewers’ comments, the descriptions for the experimental results are clarified in the revised manuscript. In addition, the related explanations and discussions are compensated and intensified in the revised manuscript. As a result, the X-ray photoelectron spectroscopy (XPS) results in Figure 2 with Table 1, the time of flight-secondary ion mass spectrometry (ToF-SIMS) results in Figure 3, and the film thickness results in Figure 5 are modified in the revised manuscript to express the experimental data more clearly as per the reviewers’ recommendations. In addition, new 13 references [14-24, 50, 51], including the published papers of the group of ‘Morent and De Geyter’, are also provided in the revised manuscript. Total figures, table, and references changed are given as follows.
Old Manuscript | Revised Manuscript |
Fig. 1 | Fig. 1 |
Fig. 2 | Fig. 2 [Modified] |
Fig. 3 | Fig. 3 [Modified] |
Fig. 4 | Fig. 4 |
Fig. 5 | Fig. 5 [Modified] |
Fig. 6 | Fig. 6 |
Fig. 7 | Fig. 7 |
Table 1. | Table 1. [Modified] |
[Ref.] 1, 2, 3, 4, 5, 6, 7, 8, 9, 10, 11, 12, 13 | [Ref.] 1, 2, 3, 4, 5, 6, 7, 8, 9, 10, 11, 12, 13 |
[Ref.] 14, 15, 16, 17, 18, 19, 20, 21, 22, 23, 24 [New] | |
[Ref.] 14, 15, 16, 17, 18, 19, 20, 21, 22, 23, 24, 25, 26, 27, 28, 29, 30, 31, 32, 33, 34, 35, 36, 37, 37, 38 | [Ref.] 25, 26, 27, 28, 29, 30, 31, 32, 33, 34, 35, 36, 37, 37, 38, 39, 40, 41, 42, 43, 44, 45, 46, 47, 48, 49 |
[Ref.] 50, 51 [New] |
II. Upon the reviewer #2’ comments
First of all, I would like to say that this manuscript on pPMMA was interesting to read and well written. I do have the following remarks on your manuscript: |
► We appreciate your kind and detailed assessment of the work presented in this paper.
1. Deposition parameters are missing: power and/or voltage. stationary deposition or moving jet? Especially since only 1 condition was tested in this paper, something which might vary from monomer to monomer in order to get the optimal deposition parameters. This limits the value of this paper. |
► The authors appreciate the in-depth comments from the reviewer. According to the reviewer’ comment, the following sentences for deposition parameters are newly provided in ‘Introduction’ and ‘Experimental and Characterizations’ in the revised manuscript.
At lines 50-52 on page 2 in Introduction:
“Our previous works have recently reported the stationary APP jets (APPJs) polymerization employing the new types of impinging technique and ATP system [25-29].”
At lines 73-95 on pages 2 and 3 in Experimental and Characterizations:
“A stationary atmospheric pressure plasma (APP) polymerization technique by using liquid monomer was conducted by an APP jets (APPJs) and aerosol-through-plasma systems mentioned in our previous work [25]. In the previous, the newly proposed guide tube and bluff body with impinging jet systems were introduced to suppress the quenching phenomenon from ambient air, thereby increasing the charged particles and plasma energies in the nucleation area [26]. The proposed APPJs with impinging technique and ATP system can produce a broadened and intense plasma discharge with large area deposition and obtain high-quality polymers, with easily controllable morphology, deposition rate, deposition size, and well-suited for a wide range of substrates. The detailed APP polymerization system, featuring the guide tube and bluff body, employed in this work, was described in the references of [25-29]. In the case of a low gas flow rate below 2500 standard cubic centimeters per minute (sccm) for main discharge gas and below 300 sccm for vaporizing or carrier monomer gas flow rates, intense or stable or broadened plasma was not produced even though the proposed guide tube and bluff body with impinging jet systems was used [28]. In this experiment, therefore, the argon (Ar) gas was a main plasma discharge gas where its purity was high (99.999%) and its flow rate was fixed at 2500 sccm. Another Ar gas was supplied to the glass bubbler for vaporizing the liquid methyl methacrylate (MMA) monomer (Sigma-Aldrich Co., St. Louis, MO, USA, Mw = 100 g·mol−1) at a fixed flow rate of 300 sccm. For plasma generation, the sinusoidal voltage with a peak value of 12 kV and a frequency of 30 kHz was fixed and applied to the proposed APP polymerization device. The used substrates were glasses and polyethylene terephthalates (PETs). The pPMMA was directly deposited on substrates at room temperature by novel APP polymerization technique. During APP polymerization experiments, the plasma jet was not moved, and as such experiments were conducted in a stationary deposition.”
2. XPS the XPS deconvolution is still missing some information (program to do it, relative sensitivity factors, FWHM constraints... ). |
► The authors appreciate the in-depth comments from the reviewer. According to the reviewer’ comment, the fitting procedure, program, relative sensitivity factors, and FWHM are newly provided in the revised manuscript.
At lines 101-118 on page 3 in Experimental and Characterizations:
“The X-ray photoelectron spectroscopy (XPS) (ESCALAB 250XI surface analysis system, Thermo Fisher Scientific, Waltham, MA, USA) was used to investigate the surface chemical compositions and atomic concentration of the pPMMA films. In the XPS measurement, the voltage and current of the monochromatic Al Kα X-ray source (hv = 1486.7 eV) was 15 kV and 20 mA, respectively. The measurement angle was 60° and the measurement depth was estimated to range from 8 to 10 nm. The measurement area was 500 μm × 500 μm and the pressure was about 10−8 Pa. The C 1s spectrum (285.0 eV) was used to calibrate the energy scale. Elements present on the deposited surfaces were identified from XPS survey scans and quantified with Thermo Avantage software (v.5.977, Waltham, MA, USA) using a Shirley background and applying the relative sensitivity factors provided by manufacturer of the instrument. The relative sensitivity factors of C 1s and O 1s were 1.0 and 2.8, respectively. For high-resolution spectra, the constant analyzer energy modes were used at 200 eV for survey scan and 50 eV pass energy for element scan, respectively. Since the pPMMA samples and substrates were insulators, we used an additional electron gun to allow for surface neutralization to adjust the charge compensation during the measurements. To curve fit the high-resolution C 1s and O 1s peaks, the deconvolution of C 1s and O 1s peaks was analyzed by the Thermo Avantage. The peaks were deconvoluted using Gaussian–Lorentzian peak shapes (constrained between 80 and 100% Gaussian) and the full-width at half maximum (FWHM) of each line shape was constrained between 2.0 and 3.0 eV.”
3. XPS results: the high res spectra of your C1s are really poor. I would recommend remeasurements with optimized parameters. In the table, only give 1 digit after the comma. |
► According to the reviewer’ comment, the high-resolution peaks including C 1s with deconvolutions of XPS spectrum of Fig. 2 are modified in the revised manuscript. In addition, all two decimal places in the XPS data of Fig. 2 and Table 1 are modified to one decimal place.
(a)
(b) (c)
Fig. 2 [Modified]
Table 1. [Modified]
Sample | Concentrations of correlative functional group (%) | |||||
C 1s | O 1s | |||||
C-C/C-H (285.0 eV) | C-O (286.8 eV) | O-C=O (289.1 eV) | C=O (531.0 eV) | C-O (533.2 eV) | ||
pPMMA | 54.9 | 29.4 | 15.7 | 68.9 | 31.1 |
4. FT-IR: it might just be optical, but the peak for OH at 3300 seems rather small compared to what you would expect for PMMA. does this suggest fragmenting? if so, do you see this is TOF SIMS and XPS? |
► The bands from 3300 cm-1 of the OH bonding group implying the oxidation of the particles induced by the quenching from ambient air were not observed thanks to using the newly proposed guide tube and bluff body systems. This suggest fragmenting. Therefore, we also found that the small negative hydroxyl ion OH- (m/z = 17) implying the oxidation of the pPMMA thin films induced by the quenching from ambient air were observed in ToF-SIMS results. According to the reviewer’ comment, the following sentences are newly provided in ‘Results and Discussion’ in the revised manuscript.
At lines 215-217 on pages 6 and 7 in Results and Discussion:
“The small negative hydroxyl ion OH- (m/z = 17) implying the oxidation of the pPMMA thin films induced by the quenching from ambient air was observed thanks to using the proposed guide tube and bluff body with impinging jet systems.”
5. TOF-SIMS: not enough attention is given to those peaks that cannot be linked to the traditional backbone of PMMA. |
► According to the reviewer’ comment, the traditional backbone of PMMA in Fig. 3 are removed and modified in the revised manuscript, and related sentences are also modified in the revised manuscript in order to avoid potential confusion.
At lines 210-219 on pages 6 and 7 in Results and Discussion:
“As shown in Figure 3a using negative-ion mode, the ions at m/z = 31, 55, 71, 85, 141, and 185 were assigned to CH3O-, C3H3O-, C3H3O2-, C4H5O2-, C8H13O2-, and C9H13O4-, respectively [42-45]. These fragment ions were arising from the pPMMA chains. The most abundant fragment ions were CH3O- and C4H5O2-, which were the monomer in which the methyl group (CH3) was removed from the ether linkage (H3C-O-C) [43,44]. The small negative hydroxyl ion OH- (m/z = 17) implying the oxidation of the pPMMA thin films induced by the quenching from ambient air was observed thanks to using the proposed guide tube and bluff body with impinging jet systems. As shown in Figure 3b using positive-ion mode, some characteristic peaks from the pPMMA polymer chain were detected.”
(a)
(b)
Fig. 3 [Modified]
6. Deposition rate is not linear, which differs from the usual pattern observed for plasma polymerization at a fixed power. How do you explain this? also 3 point measurement is poor, prefer to have 5 points for this. |
► We appreciate your kind and detailed assessment of the work presented in this paper. Under various deposition times, the deposition rates were about 0.025 μm·min-1 for 60 min deposition time and about 0.023 μm·min-1 for 90 min deposition time, which was estimated to be low. However, the deposition rates were increased significantly and maintained after 120 min deposition time at about 0.036 μm·min-1 in proposed APP polymerization technique. Therefore, according to your comment, Fig. 5 with deposition rate changes under more various times (5 point) are modified to express the experimental data more clearly and in order to avoid potential confusion in the revised manuscript, and the following sentences are modified and newly provided in the revised manuscript.
At lines 253-255 on page 9 in Results and Discussion:
“However, the deposition rates were increased significantly and maintained after 120 min deposition time at about 0.036 μm·min-1 in proposed APP polymerization technique.”
Fig. 5 [Modified]
7. There is no stability testing whatsoever. Thickness measurements for 1 day up to 3-4 weeks into aqeous solution would be essential for this paper to be published. |
► We appreciate your valuable comment. Fig. 7 shows a contact angle lower than 90, meaning that the surface is still hydrophilic. Therefore, according to three reviewers’ comment, the use of ‘hydrophobic’ are changed to ‘increase of WCA’, and the related sentences are modified in the revised manuscript in order to avoid potential confusion.
At lines 189-191 on page 5 in Results and Discussion:
“This means that our pPMMA surface becomes carbon-rich and eventually it is expected to increase the water contact angles (WCAs) after pPMMA coating [40,41].”
At lines 269-280 on page 11 in Results and Discussion:
“To calculate the surface wettability or hydrophobicity of the newly proposed polymer, WCAs of pPMMA were measured [46-49]. The WCAs analysis is a simple, rapid, and direct method to evaluate the hydrophobic or hydrophilic feature of a surface. The variation in the WCAs on the pristine (bare) substrates and pPMMA thin films grown on glass and polyethylene terephthalate (PET) substrates when using proposed APP polymerization technique after 90 min deposition is shown in Figure 7. From Fig. 7, after depositon, we can see that the WCAs of the pPMMA thin films were gradually increased for both glass and PET substrates. The results of WCA tests showed that the pPMMA thin films could increase the WCAs after deposition. The increased WCAs of the pPMMA thin films were presumably due to the cleavage of hydrophilic groups and newly formed hydrophobic groups (C-C and C-H).”
8. you do very long deposition times, are |
► Under various deposition times, the deposition rates were about 0.025 μm·min-1 for 60 min deposition time and about 0.023 μm·min-1 for 90 min deposition time, which was estimated to be low. However, the deposition rates were increased significantly and maintained after 120 min deposition time at about 0.036 μm·min-1 in proposed APP polymerization technique. Therefore, according to your comment, Fig. 5 with deposition rate changes under more various times (5 point) are modified to express the experimental data more clearly in the revised manuscript, and the following sentences are modified and newly provided in the revised manuscript.
At lines 253-255 on page 9 in Results and Discussion:
“However, the deposition rates were increased significantly and maintained after 120 min deposition time at about 0.036 μm·min-1 in proposed APP polymerization technique.”
Fig. 5 [Modified]
9. The group of Morent and De Geyter have done quite a lot of research on PMMA like coatings using plasma in the last 4 years, including APPJ polymerization, yet none of these papers are mentioned. (Look for van Vrekhem and Cools). |
► We appreciate your kind and detailed assessment of the work presented in this paper. According to the reviewer’ comment, the new 11 references [14-24], including the published papers of the group of ‘Morent and De Geyter’, are newly provided in the revised manuscript. In addition, the related explanations and discussions including introduction are compensated and intensified in the revised manuscript.
At lines 46-50 on page 2 in Introduction:
“The APP polymerization process can obtain a high-quality polymer films and nanoparticles, with easily controllable deposition rate, size, and well-suited for a wide range of substrates. Furthermore, APP reactors do not require a vacuum atmosphere and special equipment. In spite of intensive studies, only a few groups have reported successful plasma polymerization using APP [14-24].”
New References:
[14] Deynse, A. V.; Cools, P.; Leys, C.; Geyter, N. D.; Morent, R. Surface activation of polyethylene with an argon atmospheric pressure plasma jet: Influence of applied power and flow rate. Applied Surface Science 2015, 328, 269-278.
[15] Cools, P.; Sainz-Garcı´, E.; Geyter, N. D.; Nikiforov, A.; Blajan, M.; Shimizu, K.; Alba-Elı´as, F.; Leys, C.; Morent, R. Influence of DBD inlet geometry on the homogeneity of plasma-polymerized acrylic acid films: The use of a microplasma–electrode inlet configuration. Plasma Process and Polymers 2015, 12, 1153-1163.
[16] Vrekhema, S. V.; Cools, P.; Declercq, H.; Tongel, A. V.; Vercruysse, C.; Cornelissen, M.; Geyter, N. D.; Morent, R. Application of atmospheric pressure plasma on polyethylene for increased prosthesis adhesion. Thin Solid Films 2015, 596, 256-263.
[17] Morent, R.; Geyter, N. D.; Trentesaux, M.; Gengembre, L.; Dubruel, P.; Leys, C.; Payen, E. Stability study of polyacrylic acid films plasma-polymerized on polypropylene substrates at medium pressure. Applied Surface Science 2010, 257, 372-380.
[18] Geyter, N. D.; Morent, R.; Vlierberghe, S. V.; Dubruel, P.; Leys, C.; Gengembre, L.; Schacht, E.; Payen, E. Deposition of polymethyl methacrylate on polypropylene substrates using an atmospheric pressure dielectric barrier discharge. Progress in Organic Coatings 2009, 64, 230-237.
[19] Geyter, N. D.; Morent, R.; Leys, C.; Gengembre, L.; Payen, E. Treatment of polymer films with a dielectric barrier discharge in air, helium and argon at medium pressure. Surface & Coatings Technology 2007, 201, 7066-7075.
[20] Casserly, T. B.; Gleason, K. K. Effect of substrate temperature on the plasma polymerization of poly(methyl methacrylate). Chemical Vapor Deposition 2006, 12, 59-66.
[21] Kasih, T. P.; Kuroda, S.-I.; Kubota, H. Poly(methyl methacrylate) films deposited via non-equilibrium atmospheric pressure plasma polymerization using argon as working gas. Plasma Processes and Polymers 2007, 4, 648-653.
[22] Vrekhem, S. V.; Morent, R.; Geyter, N. D. Deposition of a PMMA coating with an atmospheric pressure plasma jet. J. Coat. Technol. Res. 2018, 15(4), 679-690.
[23] Cools, P.; Geyter, N. D.; Vanderleyden, E.; Barberis, F.; Dubruel, P.; Morent, R. Adhesion improvement at the PMMA bone cement-titanium implant interface using methyl methacrylate atmospheric pressure plasma polymerization. Surface & Coatings Technology 2016, 294, 201-209.
[24] Vrekhem, S. V.; Vloebergh, K.; Asadian, M.; Vercruysse, C.; Declercq, H.; Tongel, A. V.; Wilde, L. D.; Geyter, N. D.; Morent, R. Improving the surface properties of an UHMWPE shoulder implant with an atmospheric pressure plasma jet. Scientific Reports 2018, 8, 4720.
10. Finally, in my opinion, the introduction is a bit misleading. you are using terms such as electropolymerization or electrochemical, which are not common in the plasma community. Therefore, I believed that you were doing something new while it is a continuation of what has been done before. I would recommend then to rewrite the intro and abstract conform the jargon used in our community. |
► We appreciate your kind and detailed assessment of the work presented in this paper. According to the reviewer’ comment, to conform the jargon used in plasma community, the words of ‘electrochemical’ and ‘electrochemistry’ are removed in ‘Abstract’ and ‘Introduction’ in the revised manuscript. Finally, the authors again thank the reviewer’ efforts for improving the context of this paper.
At lines 14-17 on page 1 in Abstract:
“Although the polymerized methyl methacrylate (polymethyl methacrylate, PMMA) for pin hole free layer has been studied extensively in the past, little work has been done on synthesizing films of this material using atmospheric pressure plasma-assisted electropolymerization.”
At lines 33-35 on page 1 in Introduction:
“Plasma-assisted electropolymerization is a new research field of the interaction of plasma with monomer and electrolyte solution, and it uses plasma to drive the chemical reactions [1-6].”
At lines 41-43 on page 1 in Introduction:
“Among various plasma systems, by using a nonthermal atmospheric pressure plasma (APP) as a gaseous electrode, plasma-assisted electropolymerization has attracted increasing attention in the green synthesis of polymer thin films and nanomaterials.”
At lines 55-56 on page 2 in Introduction:
“In this case, the polymer films can have the different properties depending on the types of precursors and plasma conditions.”

Reviewer 3 Report
Authors of “Synthesis and Properties of Plasma-Polymerized Methyl Methacrylate via Atmospheric Pressure Plasma Polymerization Technique” proposed interesting new technology for deposition of transparent pinhole free coatings by plasma technique. The results will be inetersting for researchers working in the field of functional layers and plasma polymers. The quality of the coatings in terms of optical properties and homogeneity seems to be very high and therefore these results are very promissing. Nevertheless, although this work has satisfactory level of novelty and originality, it needs some revision and corrections.
1) First of all the quality of XPS analysis is not sufficient. The peaks are too broad due to charging effect. The fitting has high error. Either authors must remeasure the spectra or please provide std deviations for all components. It might be useful to adjust the charge compensation during measurements.
2) The binding energies chosen by authors are not fully correct. C1s The C-C component should be placed at BE=285 eV and not at 285.2 and C(O)O is at 289 and not 289.6 eV. O1 The BE of C-O can be around 533 eV but 533,7 eV is too high. It is generally the feature of bad charge compensation.
3) Page 8. Authors should correct their conclusion regarding the hydrophobic nature of coated surface, as wca=75 degrees is still hydrophilic angle and Hydrophobic property of the surface led to wca=90 or more. Thus they should correct the paragraph. However, authors must explain the increase of wca after pPMMA coating, because according to their analysis the surface has high density of hydrophilic group ( C-O, O-C=O).
There are many small grammar and spelling errors :
Munster must be with capital M. The sentence on lines 203 and 204 has missing verb.
I recommend to revise this manuscript.
Author Response
Revised manuscript number MS #Polymers-414543 entitled “Synthesis and Properties of Plasma-Polymerized Methyl Methacrylate via Atmospheric Pressure Plasma Polymerization Technique” by C.-S. Park et al.
First of all, the authors really appreciate the reviewers’ valuable comments for the paper. Based on the reviewers’ comments, the descriptions for the experimental results are clarified in the revised manuscript. In addition, the related explanations and discussions are compensated and intensified in the revised manuscript. As a result, the X-ray photoelectron spectroscopy (XPS) results in Figure 2 with Table 1, the time of flight-secondary ion mass spectrometry (ToF-SIMS) results in Figure 3, and the film thickness results in Figure 5 are modified in the revised manuscript to express the experimental data more clearly as per the reviewers’ recommendations. In addition, new 13 references [14-24, 50, 51], including the published papers of the group of ‘Morent and De Geyter’, are also provided in the revised manuscript. Total figures, table, and references changed are given as follows.
Old Manuscript | Revised Manuscript |
Fig. 1 | Fig. 1 |
Fig. 2 | Fig. 2 [Modified] |
Fig. 3 | Fig. 3 [Modified] |
Fig. 4 | Fig. 4 |
Fig. 5 | Fig. 5 [Modified] |
Fig. 6 | Fig. 6 |
Fig. 7 | Fig. 7 |
Table 1. | Table 1. [Modified] |
[Ref.] 1, 2, 3, 4, 5, 6, 7, 8, 9, 10, 11, 12, 13 | [Ref.] 1, 2, 3, 4, 5, 6, 7, 8, 9, 10, 11, 12, 13 |
[Ref.] 14, 15, 16, 17, 18, 19, 20, 21, 22, 23, 24 [New] | |
[Ref.] 14, 15, 16, 17, 18, 19, 20, 21, 22, 23, 24, 25, 26, 27, 28, 29, 30, 31, 32, 33, 34, 35, 36, 37, 37, 38 | [Ref.] 25, 26, 27, 28, 29, 30, 31, 32, 33, 34, 35, 36, 37, 37, 38, 39, 40, 41, 42, 43, 44, 45, 46, 47, 48, 49 |
[Ref.] 50, 51 [New] |
III. Upon the reviewer #3’ comments
Authors of “Synthesis and Properties of Plasma-Polymerized Methyl Methacrylate via Atmospheric Pressure Plasma Polymerization Technique” proposed interesting new technology for deposition of transparent pinhole free coatings by plasma technique. The results will be interesting for researchers working in the field of functional layers and plasma polymers. The quality of the coatings in terms of optical properties and homogeneity seems to be very high and therefore these results are very promising. Nevertheless, although this work has satisfactory level of novelty and originality, it needs some revision and corrections. |
► We appreciate your kind and detailed assessment of the work presented in this paper.
1. First of all the quality of XPS analysis is not sufficient. The peaks are too broad due to charging effect. The fitting has high error. Either authors must remeasure the spectra or please provide std deviations for all components. It might be useful to adjust the charge compensation during measurements. |
► We appreciate your kind and detailed assessment of the work presented in this paper. Since the pPMMA samples and substrates were insulators, the XPS measurement were very difficult to obtain narrow spectra due to charging effect. Therefore, we used an additional electron gun to allow for surface neutralization to adjust the charge compensation during the measurements. According to the reviewer’ comment, to clearly describe this mechanism, the fitting procedure, program, relative sensitivity factors, and FWHM are newly provided in the revised manuscript, and the high-resolution peaks with deconvolutions of XPS spectrum of Fig. 2 are modified in the revised manuscript.
At lines 101-118 on page 3 in Experimental and Characterizations:
“The X-ray photoelectron spectroscopy (XPS) (ESCALAB 250XI surface analysis system, Thermo Fisher Scientific, Waltham, MA, USA) was used to investigate the surface chemical compositions and atomic concentration of the pPMMA films. In the XPS measurement, the voltage and current of the monochromatic Al Kα X-ray source (hv = 1486.7 eV) was 15 kV and 20 mA, respectively. The measurement angle was 60° and the measurement depth was estimated to range from 8 to 10 nm. The measurement area was 500 μm × 500 μm and the pressure was about 10−8 Pa. The C 1s spectrum (285.0 eV) was used to calibrate the energy scale. Elements present on the deposited surfaces were identified from XPS survey scans and quantified with Thermo Avantage software (v.5.977, Waltham, MA, USA) using a Shirley background and applying the relative sensitivity factors provided by manufacturer of the instrument. The relative sensitivity factors of C 1s and O 1s were 1.0 and 2.8, respectively. For high-resolution spectra, the constant analyzer energy modes were used at 200 eV for survey scan and 50 eV pass energy for element scan, respectively. Since the pPMMA samples and substrates were insulators, we used an additional electron gun to allow for surface neutralization to adjust the charge compensation during the measurements. To curve fit the high-resolution C 1s and O 1s peaks, the deconvolution of C 1s and O 1s peaks was analyzed by the Thermo Avantage. The peaks were deconvoluted using Gaussian–Lorentzian peak shapes (constrained between 80 and 100% Gaussian) and the full-width at half maximum (FWHM) of each line shape was constrained between 2.0 and 3.0 eV.”
(a)
(b) (c)
Fig. 2 [Modified]
2. The binding energies chosen by authors are not fully correct. C1s The C-C component should be placed at BE=285 eV and not at 285.2 and C(O)O is at 289 and not 289.6 eV. O1 The BE of C-O can be around 533 eV but 533,7 eV is too high. It is generally the feature of bad charge compensation. |
► We appreciate your kind and detailed assessment of the work presented in this paper. The binding energies are re-calculated and modified based on your comment. In addition, according your comment, the fitting procedure, program, relative sensitivity factors, and FWHM are newly provided in the revised manuscript, and the high-resolution peaks with deconvolutions of XPS spectrum of Fig. 2 are modified in the revised manuscript.
At lines 101-118 on page 3 in Experimental and Characterizations:
“The X-ray photoelectron spectroscopy (XPS) (ESCALAB 250XI surface analysis system, Thermo Fisher Scientific, Waltham, MA, USA) was used to investigate the surface chemical compositions and atomic concentration of the pPMMA films. In the XPS measurement, the voltage and current of the monochromatic Al Kα X-ray source (hv = 1486.7 eV) was 15 kV and 20 mA, respectively. The measurement angle was 60° and the measurement depth was estimated to range from 8 to 10 nm. The measurement area was 500 μm × 500 μm and the pressure was about 10−8 Pa. The C 1s spectrum (285.0 eV) was used to calibrate the energy scale. Elements present on the deposited surfaces were identified from XPS survey scans and quantified with Thermo Avantage software (v.5.977, Waltham, MA, USA) using a Shirley background and applying the relative sensitivity factors provided by manufacturer of the instrument. The relative sensitivity factors of C 1s and O 1s were 1.0 and 2.8, respectively. For high-resolution spectra, the constant analyzer energy modes were used at 200 eV for survey scan and 50 eV pass energy for element scan, respectively. Since the pPMMA samples and substrates were insulators, we used an additional electron gun to allow for surface neutralization to adjust the charge compensation during the measurements. To curve fit the high-resolution C 1s and O 1s peaks, the deconvolution of C 1s and O 1s peaks was analyzed by the Thermo Avantage. The peaks were deconvoluted using Gaussian–Lorentzian peak shapes (constrained between 80 and 100% Gaussian) and the full-width at half maximum (FWHM) of each line shape was constrained between 2.0 and 3.0 eV.”
(a)
(b) (c)
Fig. 2 [Modified]
3. Page 8. Authors should correct their conclusion regarding the hydrophobic nature of coated surface, as wca=75 degrees is still hydrophilic angle and Hydrophobic property of the surface led to wca=90 or more. Thus they should correct the paragraph. However, authors must explain the increase of wca after pPMMA coating, because according to their analysis the surface has high density of hydrophilic group ( C-O, O-C=O). |
► The authors appreciate the in-depth comments from the reviewer. According to the reviewer’ comment, the use of ‘hydrophobic’ are changed to ‘increase of WCA’, and the related sentences are modified in the revised manuscript in order to avoid potential confusion.
At lines 189-191 on page 5 in Results and Discussion:
“This means that our pPMMA surface becomes carbon-rich and eventually it is expected to increase the water contact angles (WCAs) after pPMMA coating [40,41].”
At lines 269-280 on page 11 in Results and Discussion:
“To calculate the surface wettability or hydrophobicity of the newly proposed polymer, WCAs of pPMMA were measured [46-49]. The WCAs analysis is a simple, rapid, and direct method to evaluate the hydrophobic or hydrophilic feature of a surface. The variation in the WCAs on the pristine (bare) substrates and pPMMA thin films grown on glass and polyethylene terephthalate (PET) substrates when using proposed APP polymerization technique after 90 min deposition is shown in Figure 7. From Fig. 7, after depositon, we can see that the WCAs of the pPMMA thin films were gradually increased for both glass and PET substrates. The results of WCA tests showed that the pPMMA thin films could increase the WCAs after deposition. The increased WCAs of the pPMMA thin films were presumably due to the cleavage of hydrophilic groups and newly formed hydrophobic groups (C-C and C-H).”
4. There are many small grammar and spelling errors : Munster must be with capital M. The sentence on lines 203 and 204 has missing verb. |
► We appreciate your kind and detailed assessment of the work presented in this paper. According to the reviewer’ comment, some words, sentences, and phrases including your comments are modified to improve the English expression including grammar of our paper. In addition, typo error and sentence of manuscript were also double checked.
At lines 122-124 on page 3 in Experimental and Characterizations:
“The surface structure and composition of the pPMMA films were examined by the time of flight-secondary ion mass spectrometry (ToF-SIMS) V instrument (ION-TOF GmbH, Munster, Germany) with a bismuth primary-ion (Bi3+) gun source.”
At lines 219-221 on page 7 in Results and Discussion:
“The ions at m/z = 15, 27, 31, 39, 41, 55, 59, 69, 77, and 91 were assigned to CH3+, C2H3+, CH3O+, C3H3+, C3H5+, C4H7+, C2H3O2+, C4H5O+, C6H5+, and C7H7+, respectively.”
At lines 237-239 on page 8 in Results and Discussion:
“As shown in the SEM results of Figure 4, the pPMMA film had a deposition rate of about 0.023 μm·min-1, and had no pits and pin holes.”
5. I recommend to revise this manuscript. |
► The authors really appreciate the reviewers’ valuable comments for the paper. Based on the reviewers’ comments, the descriptions for the experimental results are clarified in the revised manuscript. In addition, the related explanations and discussions are compensated and intensified in the revised manuscript. As a result, the XPS results in Figure 2 with Table 1, the ToF-SIMS results in Figure 3, and the film thickness results in Figure 5 are modified in the revised manuscript to express the experimental data more clearly as per the reviewers’ recommendations. In addition, new 13 references [14-24, 50, 51], including the published papers of the group of ‘Morent and De Geyter’, are also provided in the revised manuscript. Finally, the authors again thank the reviewer’ efforts for improving the context of this paper.

Round 2
Reviewer 1 Report
The authors improved a lot their paper.
the referee has some questions :
- the authors claim that they have large surface area deposition with their static plasma jet. How can they get large surface area with a plasma jet ? Can they give value ?
- the results presented in the new figure 5 are very different from those from the old Figure 5. How can these date be so different ?
- The authors claim that the change in contact angle is due to the roughness of the substrate. However, they do not give any information about the roughness of the substrate, and they Don't give any information like the Wenzel relation to explain that.
Author Response
Revised manuscript number MS #Polymers-414543 entitled “Synthesis and Properties of Plasma-Polymerized Methyl Methacrylate via Atmospheric Pressure Plasma Polymerization Technique” by C.-S. Park et al.
First of all, the authors really appreciate the reviewers’ valuable comments for the paper. Based on the reviewers’ comments, the descriptions for the experimental results are clarified in the revised manuscript. In addition, the related explanations and discussions are compensated and intensified in the revised manuscript. As a result, the atomic force microscope (AFM) results and surface roughness in Figure S1 and Table S1 are newly provided in the ‘supporting information document’ and revised manuscript to express the experimental data more clearly as per the reviewers’ recommendations. In addition, new one reference [52] is also provided in the revised manuscript. Total figures, table, and references changed are given as follows.
Old Manuscript | Revised Manuscript |
Fig. 1 | Fig. 1 |
Fig. 2 | Fig. 2 |
Fig. 3 | Fig. 3 |
Fig. 4 | Fig. 4 |
Fig. 5 | Fig. 5 |
Fig. 6 | Fig. 6 |
Fig. 7 | Fig. 7 |
Table 1. | Table 1. |
Fig. S1 [New] | |
Table S1 [New] | |
[Ref.] 1-51 | [Ref.] 1-51 |
[Ref.] 52 [New] |
I. Upon the reviewer #1’ comments
The authors improved a lot their paper. The referee has some questions: |
► We appreciate your kind and detailed assessment of the work presented in this paper.
1. The authors claim that they have large surface area deposition with their static plasma jet. How can they get large surface area with a plasma jet? Can they give value? |
► We appreciate your valuable comment. In the case of conventional APPJs without a guide-tube or bluff-body, as shown in below reference figure 1, the plasma is only produced within the area of the three array jets (or bundle of 3 glass tubes) due to the directional characteristic of the streamer-like discharge. In contrast, with the proposed APPJs, the plasma produced in the nucleation region can be transited from a narrow streamer-like discharge into a broadened glow-like discharge by properly adjusting the process parameters. Importantly, when producing the broader glow-like plasma in the nucleation region, the plasma area is dramatically enlarged (about 60-fold increase) when compared with that with the narrow streamer-like plasma produced by conventional APPJs with three arrays, as shown in below reference figure 2.
According to the reviewer’ comment, the following sentences for large surface area deposition of the proposed atmospheric pressure plasma polymerization technique are newly provided in ‘Experimental and Characterizations’ in the revised manuscript.
At lines 75-80 on page 2 in Experimental and Characterizations:
“In the case of conventional APPJs without proposed guide tube and buff body, the plasma was only produced within the area of the three array jets because of the directional characteristics of the streamer-like discharges. In the case of newly proposed guide tube and bluff body with impinging jet systems, whereas, the plasma produced in the impinging region was changed to a broadened glow-like discharge, thereby increasing the plasma region about 60-fold and deposition area [26,28].”
[Previously reference figure 1: Changes in plasma images produced in nucleation when varying process parameters for proposed APPJs: Case I is conventional APPJs without proposed guide tube and buff body and Case III is proposed APPJs with guide tube and bluff body with impinging jet systems.]
[Previously reference figure 2: Comparison of the cross-sectional area of the plasma produced by conventional jets with only three array jets and proposed APPJs, where A denotes the plasma area in conventional APPJs and B denotes the plasma area in proposed APPJs.]
2. The results presented in the new figure 5 are very different from those from the old Figure 5. How can these date be so different? |
► We appreciate your valuable comment. The all data for 60, 90, and 120 min deposition times between the old and new Figure 5 were exactly the same. In new Figure 5, we additionally measured the film thickness during 150 and 180 min deposition times for obtaining the constant deposition rates in our plasma polymerization. In Figure 5, X-axis was the deposition time (min), and Y-axis was film thickness (μm). Under various deposition times, the deposition rates (thickness/deposition time) were about 0.025 μm·min-1 for 60 min deposition time and about 0.023 μm·min-1 for 90 min deposition time, which was estimated to be low. The deposition rates are low before 120 min deposition time, because it takes time to vaporize the liquid monomer for sufficiently injecting into the monomer nucleation regions in bluff body and guide tube systems. However, the deposition rates were increased significantly and maintained after 120 min deposition time at about 0.036 μm·min-1 in proposed APP polymerization technique. This means that the deposition rates remain almost constant after 120 min deposition time. Therefore, according to your comment, the following sentences are newly provided to express the experimental data more clearly and in order to avoid potential confusion in the revised manuscript.
At lines 244-245 on page 8 in Results and Discussion:
“This means that the deposition rates in our APP polymerization system remain almost constant after 120 min deposition time.”
3. The authors claim that the change in contact angle is due to the roughness of the substrate. However, they do not give any information about the roughness of the substrate, and they Don't give any information like the Wenzel relation to explain that. |
► We appreciate your kind and detailed assessment of the work presented in this paper. According to the reviewer’ comment, the surface roughness with AFM images in Figure S1 and Table S1 are newly provided in the ‘supporting information document’ for comparing the results with the contact angle, and new one reference [52] is also provided in the revised manuscript. In addition, the related explanations and discussions are compensated and intensified in the revised manuscript.
Finally, the authors again thank the reviewer’ efforts for improving the context of this paper.
At lines 267-275 on page 9 in Results and Discussion in the ‘revised manuscript’:
“In addition, we observed a change in the WCAs depending on the substrates. The surface roughness (root mean square roughness, Rq) of the pPMMA thin films on the glass substrate was 25.9 nm, whereas the Rq of the pPMMA thin films on the PET substrate was 0.6 nm in Figure S1 and Table S1. The WCA of glass sample was decreased as the Rq was increased. The changed WCAs of the pPMMA thin films on both glass and PET substrates were due to the differences in the surface roughness [50,51]. It is a typical phenomenon that it appears in hydrophilic surfaces depending on the Wenzel theory [52]. The detailed atomic force microscope images and the Rq of the pPMMA thins films surface grown on glass and PET substrates are shown in Figure S1 and Table S1.”
At lines 17-29 on pages 1 and 2 in the ‘supporting information document’:
“Figure S1 and Table S1 show the changes in two- (2D) and three-dimensional (3D) AFM images according to the root mean square roughness (Rq) and average roughness (Ra) of pPMMA film surfaces grown on glass and PET substrates when using proposed APP polymerization technique after 90 min deposition. The surface roughness of the pPMMA films was performed on a non-contact mode by Atomic Force Microscopy (Brucker, NanoWizard II, Germany) at the Korea Basic Science Institute (KBSI; Busan). All measurements were obtained under controlled room temperature. Moreover, the scanning area was 20 μm × 20 μm and scan rate was set at 1 Hz. The Bruker NanoWizard software was used for image processing and interpretation. The surface roughness (root mean square roughness, Rq) of the pPMMA thin films on the glass substrate was 25.9 nm, whereas the Rq of the pPMMA thin films on the PET substrate was 0.6 nm in Figure S1 and Table S1. The roughness of the pPMMA thin films on both glass and PET substrates was changed; this changed roughness after 90 min deposition was mainly due to differences of surface energy and Young’s modulus between various pristine substrates.”
Figure S1 [New]
Table S1 [New]
Sample | pPMMA on glass | pPMMA on PET |
Rq | 25.9 nm | 0.6 nm |
Ra | 19.9 nm | 0.4 nm |
New References:
[52] Patel, K. H.; Rawal, S. K. Exploration of wettability and optical aspects of ZnO nano thin films synthesized by radio frequency magnetron sputtering. Nanomater Nanotechnol, 2016, 6, 22.

Reviewer 2 Report
The authors have done extenxive corrections, elevating the quality of the paper.
I agree to publication as is.
Author Response
Revised manuscript number MS #Polymers-414543 entitled “Synthesis and Properties of Plasma-Polymerized Methyl Methacrylate via Atmospheric Pressure Plasma Polymerization Technique” by C.-S. Park et al.
First of all, the authors really appreciate the reviewers’ valuable comments for the paper. Based on the reviewers’ comments, the descriptions for the experimental results are clarified in the revised manuscript. In addition, the related explanations and discussions are compensated and intensified in the revised manuscript. As a result, the atomic force microscope (AFM) results and surface roughness in Figure S1 and Table S1 are newly provided in the ‘supporting information document’ and revised manuscript to express the experimental data more clearly as per the reviewers’ recommendations. In addition, new one reference [52] is also provided in the revised manuscript. Total figures, table, and references changed are given as follows.
Old Manuscript | Revised Manuscript |
Fig. 1 | Fig. 1 |
Fig. 2 | Fig. 2 |
Fig. 3 | Fig. 3 |
Fig. 4 | Fig. 4 |
Fig. 5 | Fig. 5 |
Fig. 6 | Fig. 6 |
Fig. 7 | Fig. 7 |
Table 1. | Table 1. |
Fig. S1 [New] | |
Table S1 [New] | |
[Ref.] 1-51 | [Ref.] 1-51 |
[Ref.] 52 [New] |
I. Upon the reviewer #2’ comments
The authors have done extensive corrections, elevating the quality of the paper. I agree to publication as is. |
► We appreciate your kind and detailed assessment of the work presented in this paper. Finally, the authors again thank the reviewer’ efforts for improving the context of this paper.
